# Hydrodynamic instabilities in membrane systems with current loading, Fourier analysis

**Sławomir Grzegorczyn**[1]*, **Paweł Dolibog**[1], **Iwona Dylong**[1], **Andrzej Ślęzak**[2]

**1** Department of Biophysics, Faculty of Medical Sciences in Zabrze, Medical University of Silesia, Zabrze, Poland, **2** Department of Healt Sciences and Physiotherapy, Collegium Medicum, Jan Dlugosz University, Częstochowa, Poland

* grzegorczyn@sum.edu.pl

## Abstract

Time-current characteristics for membrane systems with bacterial cellulose membranes located in horizontal plane and with NaCl solutions, indicate a stable formation of concentration boundary layers near the membrane for the configuration with a solution of lower concentration and lower density above the membrane (A). In turn, for the configuration with a higher concentration above the membrane (B) and a sufficiently large initial concentration quotient on the membrane ($(C_h/C_l)_o$) current pulsations are observed over time, resulting from hydrodynamic instabilities occurring in the vicinity of the membrane. The increase of $(C_h/C_l)_o$ in configuration B causes an increase in the frequency of current pulsations and a change in their amplitudes. Furthermore, significant differences were observed between the types of the temporal changes in membrane currents in both configurations. The currents measured in steady states after 24 hours show differences between A and B configurations for $(C_h/C_l)_o > 500$. Fast Fourier Transform (FFT) used to analyze the hydrodynamic instabilities in the range of observed pulsations of the measured currents shows that the average signal power of currents in the frequency range from 0.05 to 1 min$^{-1}$, depends non-linearly on the initial concentration quotient on the membrane, showing a maximum for the $(C_h/C_l)_o$ equal to 2500. In turn, the Short Time Fourier Transform (STFT) applied to the current signal as well as to the current difference signal (time lag equal to 1 min.) showed better resolution of the hydrodynamic instability analysis for the current difference signal. The increase of $(C_h/C_l)_o$ in the membrane system causes a gradual increase in STFT amplitudes towards longer times and higher frequencies. In turn, significant activity of hydrodynamic instabilities was observed in the first 50 min of current measurements for $(C_h/C_l)_o > 2500$, followed by suppression of these instabilities.

**Data availability statement:** All relevant data are within the manuscript and its Supporting information files.

**Funding:** Iwona Dylong and Sławomir Grzegorczyn BNW-1-048/K/3/I Medical University of Silesia https://sum.edu.pl. The funders had no role in study design, data collection and analysis, decision to publish, or preparation of the manuscript.

**Competing interests:** The authors have declared that no competing interests exist.

## Introduction

Bacterial cellulose membranes (*Biofill*), due to their structure characterized by high porosity and degree of hydration, have good transport properties for both electrolytes and non-electrolyte substances. For this reason, they have found applications in solving various technical [1–3] as well as medical [4,5] problems. Due to their impermeability to bacteria and, at the same time, easy exchange of water and small molecules, thin sheets of bacterial cellulose are used as dressings for hard-to-heal wounds [6]. Bacterial cellulose is an interesting and increasingly popular material due to its significantly smaller cellulose fibers and fewer other components compared to plant cellulose. For this reason, bacterial cellulose has found numerous applications in medicine [7] and technology [8,9]. The membrane itself constitutes a barrier (physical, structural, electrical) to the flow of solutions between the separated compartments, which allows for controlled transport of substances in single- and multi-membrane systems. Due to their flow-limiting properties and the ability to control the fluxes of substances, membranes have found wide practical applications, including seawater desalination [10,11], wastewater treatment in both industrial and municipal processes [12], electricity generation [10,13,14], and others. In each of these applications, membrane concentration polarization is important, and research is currently underway to limit its impact on membrane transport.

One of the important models describing the transport of non-electrolyte substances and electrolytes through membranes is the Kedem-Katchalsky (KK) model [15]. This model assumes homogeneous solutions on both sides of the membrane, most often ensured by mechanical stirring of solutions. The essence of the model is the equations

$$J_v = L_p \left( \Delta P - \sum_{j=1}^{n} \sigma_j \cdot \Delta \pi_j + \beta \cdot I \right)$$

(1)

$$J_s = \overline{C}_s \left( 1 - \sigma_s \right) J_v + \sum_{j=1}^{n} \omega_{sj} \Delta \pi_j + \frac{t_s}{z_s F} \cdot I$$

(2)

$$I = -\kappa \cdot \beta \cdot J_v + \kappa \cdot \sum_{j=1}^{n} \frac{t_j}{z_j F} \frac{RT}{C_j} \Delta C_j + \kappa \cdot E$$

(3)

where $J_v$ and $J_s$ are the volume and ion fluxes ($s$ – indexes for suitable ions, $n$ number of ions in solution), $I$ is the density current through the membrane, $\Delta P$ is the difference of mechanical pressure through the membrane, $\Delta \pi_j$ is the osmotic pressure difference through the membrane for $j$ solute, $\overline{C}_s = \left( C_h - C_l \right) \left[ \ln \left( C_h C_l^{-1} \right) \right]^{-1}$ is an average $s$ solute concentration in the membrane and $E = \frac{\Delta \overline{\mu}}{z_s F}$ is the difference of electrical potential on the membrane. Besides, $C_h$ and $C_l$ ($C_h > C_l$) are the solute concentrations in the chambers at the initial moment, $L_p$, $\sigma_s$ and $\omega_{sj}$ are hydraulic

permeability, reflection and solute permeability coefficients for membrane suitably. $\beta$, $t_s$ and $\kappa$ are electroosmotic coefficient, transference number of ions $s$ and conductivity of the membrane suitably. Besides $F$, $R$ and $T$ are the Faraday number, gas constant and absolute temperature suitably, $\overline{\mu}$ is the electrochemical potential of solution and $z_s$ is the valence of ion s.

In practice, the phenomenon of concentration polarization of the membrane occurs, which causes heterogeneity of solutions near the membrane surface. This requires extending the KK model to the case of heterogeneous solutions [16]. One possible extension of the KK model is the model of solutions layers in membrane chambers using the diffusion equation for solutions transport in chambers. The diffusion equation can be shown as

$$\frac{\partial C_s}{\partial t} = -\frac{\partial J_s}{\partial x} = D_s \cdot \frac{\partial^2 C_s}{\partial x^2}$$

(4)

where $D_s$ is the diffusion coefficient of solute $s$ in aqueous solutions. As was previously mentioned, one of the important factors influencing the transport of substances through the barrier created by the membrane is the phenomenon of concentration polarization of the membrane [16,17]. This phenomenon is associated with the appearance of concentration gradients of transported substances, perpendicular to the membrane surface and significantly decreasing the fluxes of transported substances [17,18]. The associated layers with concentration gradients in the vicinity of membranes, called concentration boundary layers (CBLs) [19,20], are often investigated for potential reduction of concentration polarization of membrane [21,22] and thus decreasing their impact on membrane transport. Another phenomenon that may appear in CBLs are hydrodynamic instabilities [18], which may be caused by solution density gradients directed opposite to the gravitational field vector. Other causes of hydrodynamic instabilities related to density disturbances may be large currents flowing through solutions [23,24] or chemical reactions occurring in solutions [25–27]. Disturbance of the density near the membrane by various factors can lead to the blurring of CBLs and causing a significant reduction in the concentration polarization of the membrane and thus an increase in solutes fluxes through the membrane. Density gradients near membrane can be caused by temperature gradients or by concentration gradients of substances whose densities depend on the concentration of the substance. At sufficiently large density gradients, additional convective fluid motions can occur, leading to hydrodynamic instabilities near the membrane [28–30]. A parameter that is crucial in the boundary conditions where hydrodynamic instabilities arise in CBLs is the critical Rayleigh number. The Rayleigh number itself can be represented as [31]

$$R_a = \frac{g \cdot \frac{\partial \rho}{\partial C} \cdot \Delta C \cdot \delta^3}{\rho \cdot \nu \cdot D_s}$$

(5)

where $g$ is the gravitational acceleration, $\Delta C$ is the difference of concentrations in CBL, $\delta$ is the thickness of CBL, $\rho$ and $\nu$ are the density and the kinematic viscosity coefficients of solution suitably, $D_s$ is the diffusion coefficient for solute $s$ in solution. Critical value of the Rayleigh number depends on the type of system in which the displacements of substances are considered [32]. The value of the Rayleigh number allows determining the stability conditions of solutions in CBLs and the type of flows observed in the chambers of the membrane system. In the case of membranes, when the membrane surface can be considered as a fixed rigid CBL border, while the second CBL boundary is blurred, the critical value of the Rayleigh number is taken as 1100.6 [32]. Below this value, the solution flow through the CBL is laminar (diffusive creation of CBLs), whereas after exceeding the critical value of the Rayleigh number, the diffusive flows are disturbed by turbulent displacements of solutions in the CBLs (hydrodynamic instabilities in CBLs).

The method we use to study the dynamics of CBL evolution in membrane systems is a method based on the electrodes that are reversible with respect to the electrolyte (e.g., Ag|AgCl electrodes for chloride solutions) [20,33]. It allows the measurement of voltages between electrodes immersed directly in solutions, using a millivoltmeter with high internal

resistance. The voltage between the electrodes in the membrane system, immersed directly in the solutions, can be written as [33,34]

$$\Delta\psi = -\frac{2RT}{F}\left[(t_+^m - t_+) \cdot ln\left(\frac{\gamma_h^m C_h^m}{\gamma_l^m C_l^m}\right) + t_+ \cdot ln\left(\frac{\gamma_h^{el} C_h^{el}}{\gamma_l^{el} C_l^{el}}\right)\right]$$

(6)

where: $\gamma_i^m C_i^m$, $\gamma_i^{el} C_i^{el}$ are the products of ion activity coefficients and concentrations at membrane ($m$) and electrode ($el$) surfaces suitably (indices $i$ refer to chambers $h$ or $l$), $t_+^m$ and $t_+$ are the apparent transference numbers for Na$^+$ ions in membrane and in chamber suitably.

Time characteristics of voltages in membrane systems enable the study of diffusional CBLs reconstruction and the determination of the conditions of appearance and the type of instability in CBLs [16,28,35]. Measurement of the time characteristics of membrane currents, with the use of an electrical meter with significantly lower resistance should allow for the expansion of the scope of these studies. In this case, electrical measurements in a membrane system would occur with significantly greater charge transport through the membrane and CBLs, i.e., under conditions of significantly higher current load. The measured membrane current can be presented in the form [36]

$$I = \frac{\Delta\psi}{R_A + R_{els} + R_m + R_{CBLs} + R_w}$$

(7)

As can be seen from the equation (7), the membrane current depends on the voltage ($\Delta\psi$), generated between the electrodes immersed directly in the solutions, as well as on the resistances of the membrane itself ($R_m$) and the solutions in CBLs (between electrodes and membrane – $R_{CBLs}$ ), as well as the resistances of the electrodes ($R_{els}$), the connecting wires ($R_w$) and the internal resistance of the nanoammeter ($R_A$). For this reason, the dynamics of voltage changes over time in membrane systems with negligible currents may differ significantly from the dynamics of membrane currents measured under different concentration conditions in the membrane system. This will allow the analysis of transport conditions in membrane systems with large charge transports through the membrane.

In our case, the main direction of interest was focused on hydrodynamic instabilities in CBLs and their characterization by time dependence of currents in membrane systems with membrane in the horizontal plane and mainly in the configuration in which the higher density solution was located above the membrane (configuration B). The stability conditions of solutions in CBLs are determined by the density gradients in solutions (caused by concentration gradients), which influence the Rayleigh number defined by the equation (5). Their nature, reflected in the time characteristics of membrane currents for different initial conditions, should depend on these conditions, i.e., the initial solutes concentrations quotient on both sides of the membrane. Due to the fact that the basic manifestation of hydrodynamic instabilities in the membrane system are pulsations of the measured electrical parameters [37,38], we decided to use the method of analysis of the time-varying current signals – the Fast Fourier Transform (FFT). The analysis was carried out for time intervals in which hydrodynamic instabilities appear, visible as pulsations of the measured currents. The results of the analysis are presented using the appropriate transformation spectra of these time intervals. To parametrized the Fourier analysis of the observed processes for different initial conditions, the average signal power of the Fourier spectrum was determined in suitable frequency ranges.

An important problem related to the Fourier analysis of hydrodynamic instabilities is whether specific frequencies of concentration changes in CBLs (and connected with them pulsations of measured currents in the membrane system) can be identified or whether the Fourier spectra would indicate the random nature of these processes. In addition, Fourier analysis should show the main frequency ranges of observed pulsations of measured currents and the distribution of the signal power vs frequency after the transform. Another problem is to determine how current pulsations (their frequency distribution) behave over time. To determine this, the Short Time Fourier Transform (STFT) was applied to

the measured currents. In turn, due to the decreasing trend line in the current signals, which may significantly interfere with the interpretation of hydrodynamic instabilities, the current signals were subjected to a transformation consisting in replacing the currents with their current differences separated in time by 1 minute. This transform aims to remove the decreasing trend line and leave only the pulsations related to hydrodynamic instabilities in the new signal. The signals obtained in this way were also subjected to STFT in order to visualize the distribution of Fourier amplitudes in the time-frequency plane. In addition, using FFT for the new differential signals of currents, the average signal powers after the transformation were calculated in twenty-minute intervals, with interval centers at selected times for different initial conditions. The average signal powers after Fourier transformation should indicate the intensity of hydrodynamic instabilities in selected time intervals. These procedures should allow for better description of the dynamics of transport processes in the membrane systems with large currents and to characterize hydrodynamic instabilities in chambers of the membrane system.

## Materials and methods

A two-chamber system with a bacterial cellulose membrane (*Biofill* membrane, Fibrocel Productos Biotechnologicos Ltd. Curitiba, Brazil) and a mechanical solution stirring system were used to measure currents in the membrane system. The membrane systems consisted of two cylindrical chambers with volumes $200\,cm^3$ each, while the surface of the membrane separating chambers was S = $6.1 cm^2$. The transport parameters of the bacterial cellulose membrane for NaCl solutions: $L_p$ = $0.5 \cdot 10^{-11}$ $m^3 N^{-1} s^{-1}$, $\sigma_s$ = 0.06 and $\omega_s$ = $14.3 \cdot 10^{-10}$ $mol N^{-1} s^{-1}$ were determined experimentally [38]. A nanoammeter with an internal resistance 0.9 MΩ, a range 100 nA, and a resolution 0.1 nA (Meratronik U276) was connected to a computer with software for recording temporal changes in membrane currents. Ag|AgCl electrodes were used (made by repeatedly immersing purified silver wire in molten AgCl and then thermally stabilized). The lengths of prepared electrodes were 14.0 mm, diameter 1.0 mm, while their electrical resistance depended only slightly on the solution concentration ($R_{els}$ = 0.95MΩ). In turn, the membrane resistance, calculated for example for NaCl concentration of 0.01 mol $m^{-3}$, is about 5.2 kΩ [36] and is negligibly small compared to the resistance of the electrodes and the internal resistance of the nanoammeter. The entire measurement system was placed in a grounded and thermally stabilized metal enclosure, protecting it from the influence of external electric and electromagnetic fields. The scheme of the measurement system is shown in Fig 1, in two configurations, A and B, respectively.

During measurements of currents, the measuring system with the membrane in the horizontal plane, had two configurations: with a higher-density solution below the membrane (configuration A) and with a higher-density solution above the membrane (configuration B). The bacterial cellulose membrane (*Biofill*), and aqueous NaCl solutions of various concentrations were used. For NaCl solutions, the higher the NaCl concentration, the higher the solution density. This allows to change the conditions for CBLs formation over time by selecting appropriate solution concentrations at the initial moment and thus control the conditions for the development of hydrodynamic instabilities in the system. The initial lower concentration in one chamber was set at $10^{-5}$ mol/l NaCl, while in the second chamber the concentration of NaCl aqueous solution at the beginning of experiment was varied from $5 \cdot 10^{-5}$ mol/l to $7.5 \cdot 10^{-2}$ mol/l. Before the measurement, the solutions in chambers were stirred (for a maximum of 2 minutes) to ensure homogeneity of solutions. The moment of turning off the stirring of the solutions was the starting point for the measurement of membrane current. The quotient of concentrations in the chambers $(C_h/C_l)_o$ was chosen as a parameter of the initial conditions for the membrane system, where $C_h$ and $C_l$ are bulk solutions, homogeneous in both chambers at the beginning of the experiment.

The current values were measured every 4 seconds for 6 hours, and then, to determine the steady state of the membrane system, the current was additionally measured 24 hours after the turning off mechanical stirring of the solutions. The error in the preparation of NaCl solution concentrations did not exceed 2%. The standard error of measurements of the initial current do not exceeded 5%, while the standard error of measurements of currents in steady states after 24h

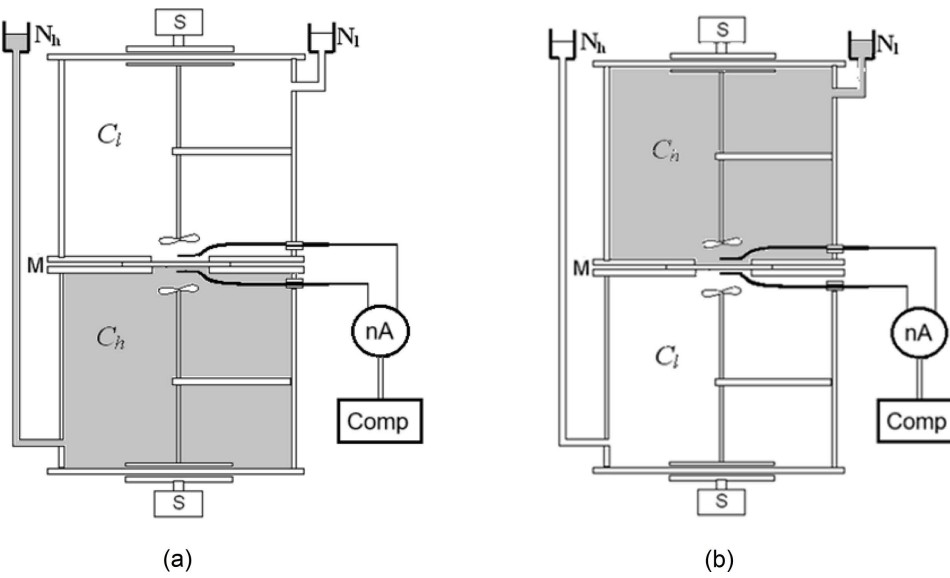

**Fig 1. Scheme of the measuring system with a bacterial cellulose membrane (M) in configurations A (a) and B (b), with aqueous NaCl solutions with initial concentrations $C_h > C_l$, S is the motor of the solution stirring system, nA is the nanoammeter and Comp is a computer.**

do not exceeded 12%. The estimations of the measurement error of the current intensities in the initial and in the steady states were performed five times for the selected initial quotients of NaCl concentrations in the chambers $(C_h/C_l)_o$.

The current as a function of time, obtained for configuration B of the membrane system and characterized by current pulsations related to hydrodynamic instabilities, was analyzed using the Fast Fourier Transform (FFT). The Origin Pro 2024 was used for Fourier analysis and graphical presentation of obtained results. FFT analysis was performed in intervals in which current pulsations were observed: from 50 to 250 minutes, and in smaller 20-minute intervals with interval centers at 20, 100, 150, 200, and 290 minutes.

We chose the MSA power density as a parameter characterizing the frequency spectrum of the FFT transform. The method used in Origin Pro is Periodogram, which estimates the power from the amplitude of the Fourier transformed data, and the Power density is defined as

$$P_{MSA} = \frac{Re^2 + Im^2}{n^2} \tag{8}$$

where Re and Im are the real and imaginary parts of transformed data and $n$ is the length of the input sequence ($P_{MSA}$ = $(Amplitude)^2$). Moreover, due to the fact that the main frequencies with non-zero signal power occurred in the frequency range up to $1\,min^{-1}$, a parameter characterizing the average signal power in the range from 0.05 to $1\,min^{-1}$ was also calculated according to the equation

$$P_{AV} = \frac{\int_{0.05\,min^{-1}}^{1\,min^{-1}} \{P_{MSA}(f)\}\ df}{(1\,min^{-1} - 0.05\,min^{-1})} \tag{9}$$

where: $P_{MSA}(f)$ – signal power in the frequency domain, $f$ – signal frequency, $P_{AV}$ – average signal power in the selected frequency range. The lower limit of integration was adopted due to the high amplitude of the signal power for low

frequency values associated with the continuous decrease in current over time caused by the gradual disappearance of the concentration forces on the membrane and in the CBLs.

Furthermore, to visually determine the frequency distribution (their changes) during the experiment, the current signals were transformed by the Short Time Fourier Transform (STFT) using Origin Pro 2024. The STFT parameters in Origin Pro 2024 application have been set to: Sampling interval = 4 s; FFT Length = 512; Window Length = 301 (20 minute window of analysis); Overlap = 150; Window type: Rectangle; Window correction: Amplitude; Option: Amplitude result. We focus mainly on the analysis of hydrodynamic instabilities in the membrane system. Therefore, to eliminate the systematic decrease of membrane currents over time resulting from the electrolyte transport through the membrane, we used differential method. This method consists in replacing the elements of the measurement series ($I(t_i)$) with the differences between the points of the time series that are separated in time by the same interval ($\Delta I(t_i)$). We used the algorithm: $\Delta I(t_i) = I(t_i) - I(t_i + 1\,\text{min.})$. The time interval between subtracted currents amounts one minute because no pulsations with a period of less than 1 minute were observed. For smaller time intervals, the graphs are similar but with a smaller amplitude and a larger number of high-frequency components not observed in the current signal.

## Results and discussion

Figure 2 presents the time characteristics of membrane currents for different $(C_h/C_l)_o$ in the membrane system with a bacterial cellulose membrane. As for the nature of the changes, the curves illustrating the time characteristics of the membrane currents, presented in Fig 2, are similar to the curves illustrating the time characteristics of the voltages between electrodes immersed into solutions. The differences between the time characteristics of membrane voltages and currents may be related to different values of the current density flowing through the membrane during measurements and changes in resistances in the membrane system. During measurement of time characteristics of voltages between electrodes in the membrane system with a millivoltmeter, the maximal current was lower than 1.5 nA. In turn, for time characteristics of currents in the membrane system with a nanoammeter, the measured currents were several dozen times greater, but did not exceed 100 nA.

Turning off the mechanical stirring of solutions causes the value of membrane currents to decrease over time, which is associated with the evolution in time of CBLs. The greatest changes in current are observed in the first few minutes after turning off mechanical stirring of solutions in the membrane system, after which the rate of change in current becomes slower. This results from concentration changes in the membrane and in CBLs as a result of ion diffusion through and near the membrane. For $(C_h/C_l)_o \leq 100$, the time characteristics of the membrane current are similar in both membrane system configurations. In turn, the time characteristics of membrane currents for configurations A and B and $(C_h/C_l)_o \geq 100$ differ significantly. A characteristic feature of the temporal changes in the currents observed in configuration A is the gradual decrease of current over time, which is the effect of diffusional evolution of CBLs over time. For configuration B and for $(C_h/C_l)_o \geq 100$, current pulsations appear due to hydrodynamic instabilities in CBLs. The time of current pulsation is longer than 300 minutes, with the current pulsation amplitude decreasing systematically during this time. For $(C_h/C_l)_o > 1000$, current pulsations are characterized by a short pulsation period of current with initially large amplitude, decreasing over time. This may be related to the gradual disappearance of thermodynamic forces supporting hydrodynamic instabilities (density gradients in CBLs directed oppositely to the gravitational field vector). The short period of current pulsation (approximately 2–3 minutes) indicates quick fluid movements in the CBLs (and near the electrode surfaces), resulting from the large enough density gradients. These processes, leading to fast stirring of layers of different density, contribute to the disappearance of the thermodynamic forces, and thus to a reduction in the mass exchanged between the layers. As a result, concentration changes on the electrode surfaces are smaller and, consequently, the amplitude of pulsations of currents flowing through the membrane is reduced.

Figure 3 shows the concentration characteristics of membrane currents for bacterial cellulose membranes, measured in steady state 24 hours after turning off the mechanical stirring of solutions. The currents in the membrane system during

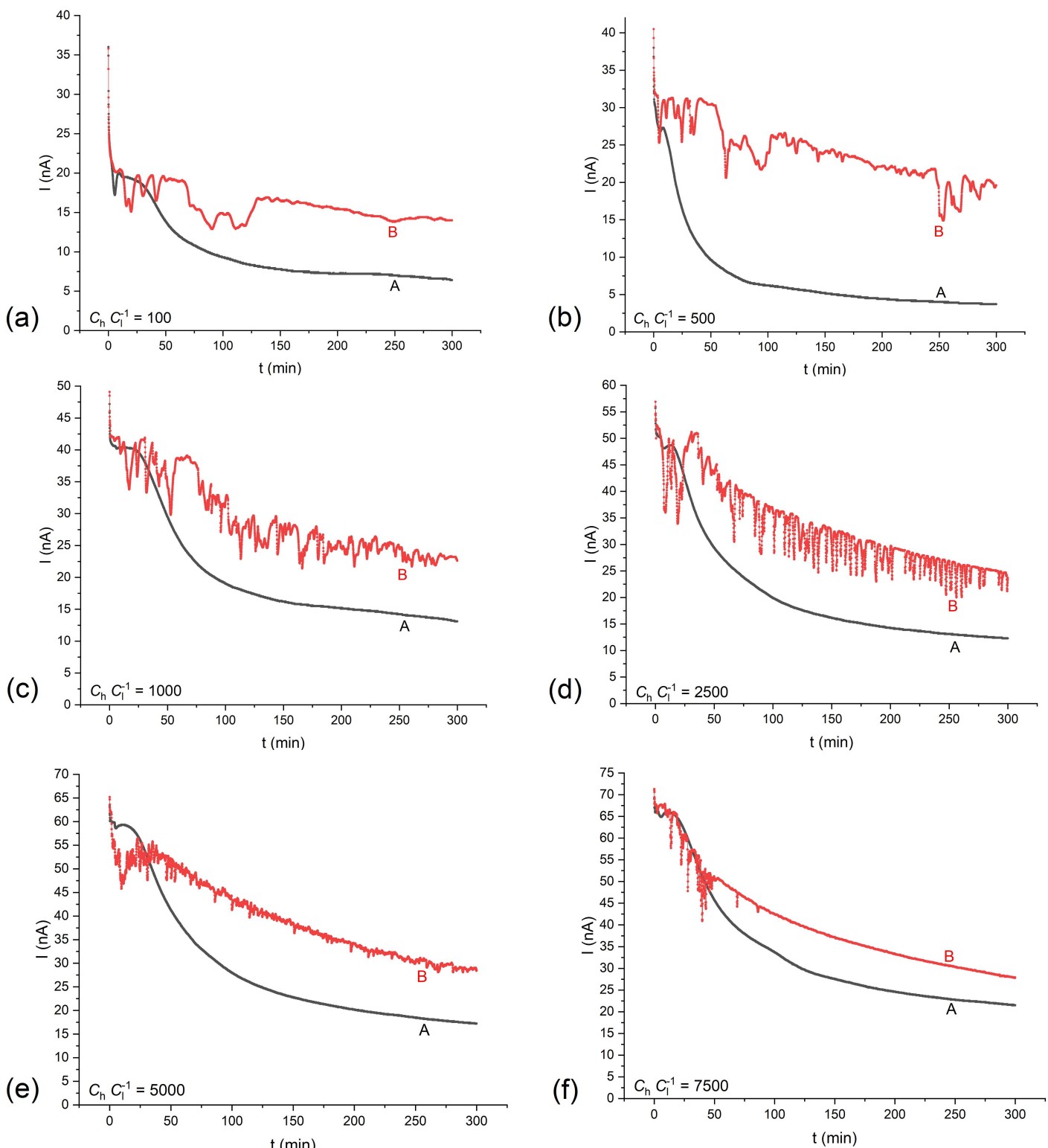

**Fig 2. Time characteristics of membrane currents (*I*) for the membrane systems with bacterial cellulose membranes and electrodes at a distance of 5 mm from both membrane surfaces, for configurations A and B, for $(C_h/C_l)_o$ equal to: 100 (a), 500 (b), 1000 (c), 2500 (d), 5000 (e) and 7500 (f).**

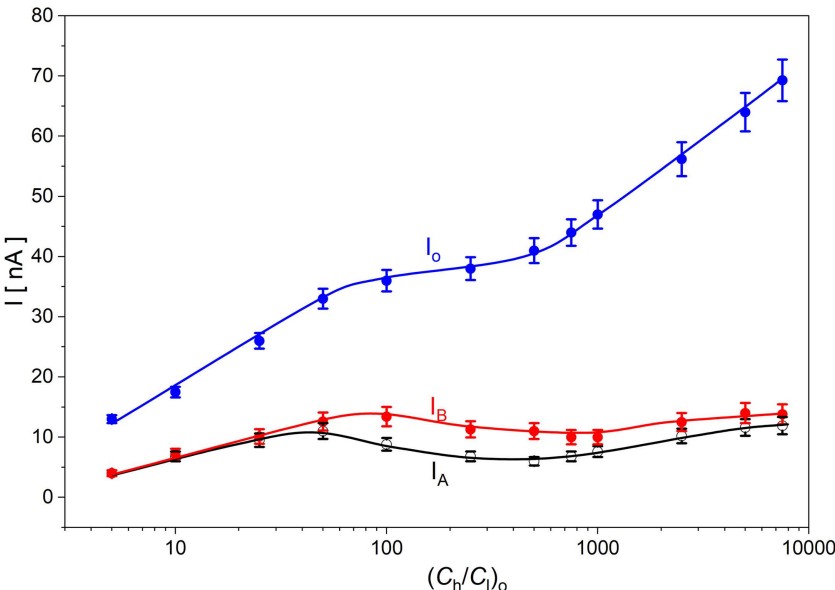

**Fig 3. Membrane currents as functions of $(C_h/C_l)_o$ for the membrane systems with bacterial cellulose membranes: $I_o$ - measured at the initial moment during mechanical stirring of solutions, $I_A$ and $I_B$ – measured in steady states for configurations A and B, respectively.**

mechanical stirring of solutions are marked as $I_o$. In turn, the currents in steady states were marked for configuration A as $I_A$ and for configuration B as $I_B$. The currents for $I_o$ correspond to the average of currents for both configurations of the membrane system during mechanical stirring of solutions.

Figure 3 shows that the increase in $(C_h/C_l)_o$ is the reason for the nonlinear increase in the current flowing through the membrane during mechanical stirring of solutions in the chambers, to a value of about 70 nA for $(C_h/C_l)_o = 7500$. Higher $(C_h/C_l)_o$ values at the initial moment mean higher concentration gradients on the membrane at the beginning and appearing over time in the CBLs, which is associated with higher ion fluxes diffusing through the membrane and through the CBLs. These factors contribute to the higher observed currents in the membrane system. Moreover, for $(C_h/C_l)_o$ lower than 25 there are no significant differences between the steady-state membrane currents for configurations A and B. An increase in $(C_h/C_l)_o$ above 25 causes the steady-state currents for configuration A to be lower than for configuration B. As it results from the analysis of the time- characteristics of currents in membrane systems, current pulsations are observed for $(C_h/C_l)_o$ greater than or equal to 100. However, differences in the steady-state currents ($I_B > I_A$) appear for $(C_h/C_l)_o$ greater than 50. The reason for the differentiation is the hydrodynamic instabilities caused by gravity. However, for lower thermo-dynamic forces in CBLs they are not observed in the form of current pulsations, but can be seen in the steady states for both configurations of the membrane system. In both configurations, the CBLs rebuild by diffusion, while hydrodynamic instabilities occurring in configuration B cause the CBLs to "blur" and are responsible for a slower increase in CBLs thicknesses than in configuration A. An increase in $(C_h/C_l)_o$ increases the difference between the steady-state currents in both configurations. The maximal difference between steady states of currents is approximately for $(C_h/C_l)_o = 250$. A further increase in $(C_h/C_l)_o$ causes a decrease in the differences between the steady states currents in configurations A and B. This means a reduction in the intensity of the processes generating hydrodynamic instabilities in the areas adjacent to the membrane, which determine the thicknesses of the CBLs.

FFT and STFT were used to analyze the current pulsations over time caused by hydrodynamic instabilities in the CBLs. The signal in the range of observed current pulsations, ranging from 50 to 250 minutes, was analyzed by means of FFT and the results are shown in Fig 4.

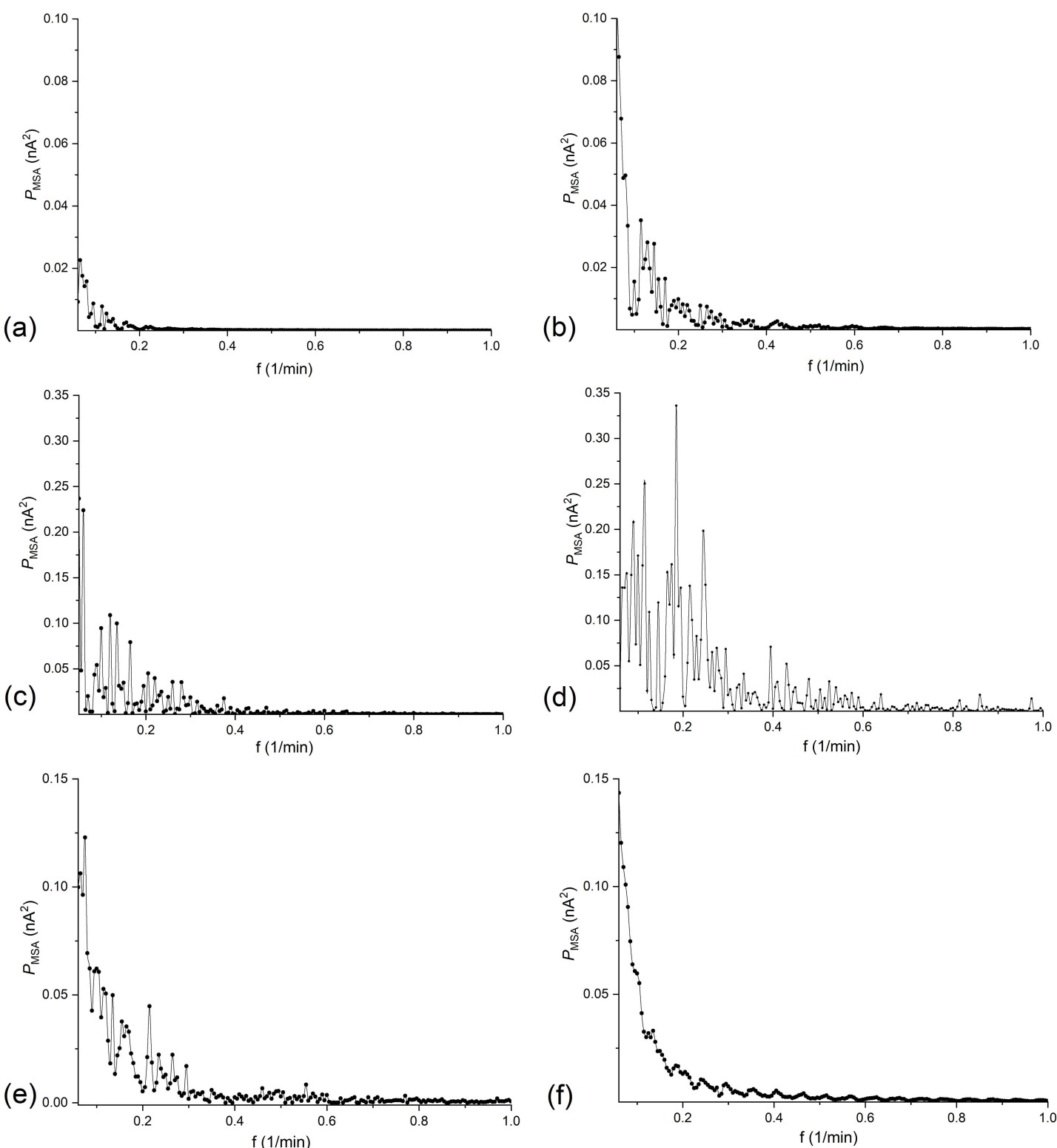

**Fig 4. Dependence of the signal power ($P_{MSA}$) on the frequency, obtained as a result of FFT in time interval 50-250 min., for configuration B of the membrane system and for $(C_h/C_l)_o$ equal to: 100 (a), 500 (b), 1000 (c), 2500 (d), 5000 (e) and 7500 (f).**

As can be seen from graphs 4 a-f, the main component of the signal power is in the low frequency range up to $1\,min^{-1}$. This is consistent with the graphs for configuration B presented in Fig 3, which show that the current pulsation periods are usually in the range of few to several dozen minutes. The main feature of the FFT spectra, is that with the increase in $(C_h/C_l)_o$, the powers in the observed range of frequency initially increase and above $(C_h/C_l)_o = 2500$ they decrease. Moreover, the frequency range with high power values also shifts towards higher frequencies with increasing $(C_h/C_l)_o$ and above $(C_h/C_l)_o = 5000$ towards lower ones.

In turn, the results of the STFT analysis of current signal are shown in Fig 5. The parameters specified for STFT in Origin Pro 2024: Sampling interval = 4s; FFT Length = 512; Window Length = 301 (20 minute window of analysis); Overlap = 150; Window type: Rectangle; Window correction: Amplitude; Option: Amplitude result.

The graphs in Fig 5 show that large STFT amplitudes are observed in the very low frequency range, which is mainly related to the decreasing trend line of currents connected with the rebuilding of CBLs in the membrane system. As time passes, the amplitudes in the very low frequency range decrease, which is associated with the decreasing rate of change of the current over time. Moreover, it can be observed that the emerging regions of higher amplitudes expand towards longer times and higher frequencies with increasing $(C_h/C_l)_o$. For this reason, to reduce the influence of the decreasing trend line of currents on the observed STFT characteristics, a transformation of the current signals was performed, replacing the time series of membrane current with a time series of membrane current differences shifted mutually by 1 minute. This procedure should remove the decreasing trend line and leave the membrane current pulsations directly related to the hydrodynamic instabilities. The effect of this transformation for different $(C_h/C_l)_o$ is shown in Fig 7, and the corresponding STFT of these time characteristics are shown in Fig 9.

The main frequency ranges where the power peaks were significantly higher than at other frequencies were the frequency ranges up to $0.6\,min^{-1}$. For this reason, in order to capture and better present these relationships, the average power in the range of observed signal powers, i.e., from 0.05 to $1\,min^{-1}$, was adopted as the parameter. The lower range was assumed due to the longest observed periods of current pulsation in the time characteristics of currents in the membrane system. The upper range was assumed as a cut-off from the frequency range with negligible signal powers. The signal power averaging range corresponds to the borders of pulsation periods of membrane currents: 20 min and 1 min, respectively. The average power was calculated based on the formula (9). Moreover, the averaging limit at low frequencies was chosen due to the large signal component at very low frequencies resulting from the gradual decrease in membrane current over time. Figure 6 shows the dependence of the average power in the frequency range from 0.05 to $1\,min^{-1}$ on $(C_h/C_l)_o$ in selected time-intervals for the bacterial cellulose membrane and aqueous NaCl solutions.

As can be seen from Fig 7, for both large and small time intervals, the average signal power shows a maximum for $(C_h/C_l)_o = 2500$. Moreover, Fig 7b shows that as the center of the 20-minute interval is moved towards a longer time, the average signal power in the interval decreases, while in ranges 100 and 150 min. average powers are similar. This can be related to similar intensities of hydrodynamic instabilities in the middle time intervals of observed pulsations and then a gradual reduction in the intensity of hydrodynamic instabilities. The CBLs formation depends on two main processes: solute transport through the membrane and solute diffusion through CBLs. These processes lead to an increase in the CBLs thickness over time and possibility of appearance of hydrodynamic instability in the CBLs. The hydrodynamic instabilities observed only in configuration B cause a gradual blurring of the outer boundaries of the CBLs. The first process leads to a gradual decrease in the current in the membrane system while the second leads to the appearance of pulsations in time characteristics of measured currents. The graphs obtained as a result of the procedure of replacing membrane currents by the differences between currents shifted in time by 1 minute ($\Delta I$) for different $(C_h/C_l)_o$ are presented in Fig 7.

As can be seen from Fig 7, the pulsations in the graphs correspond to the pulsations of the time dependencies of the currents in configuration B shown in Fig 2. Figure 7 shows that the decreasing trend of the current over time has been eliminated, which was the goal of used algorithm. The lack of a trend line in time characteristics should reduce very low-frequency components in the FFT spectrum. Figures 8 (FFT analysis of differences of currents the time range from

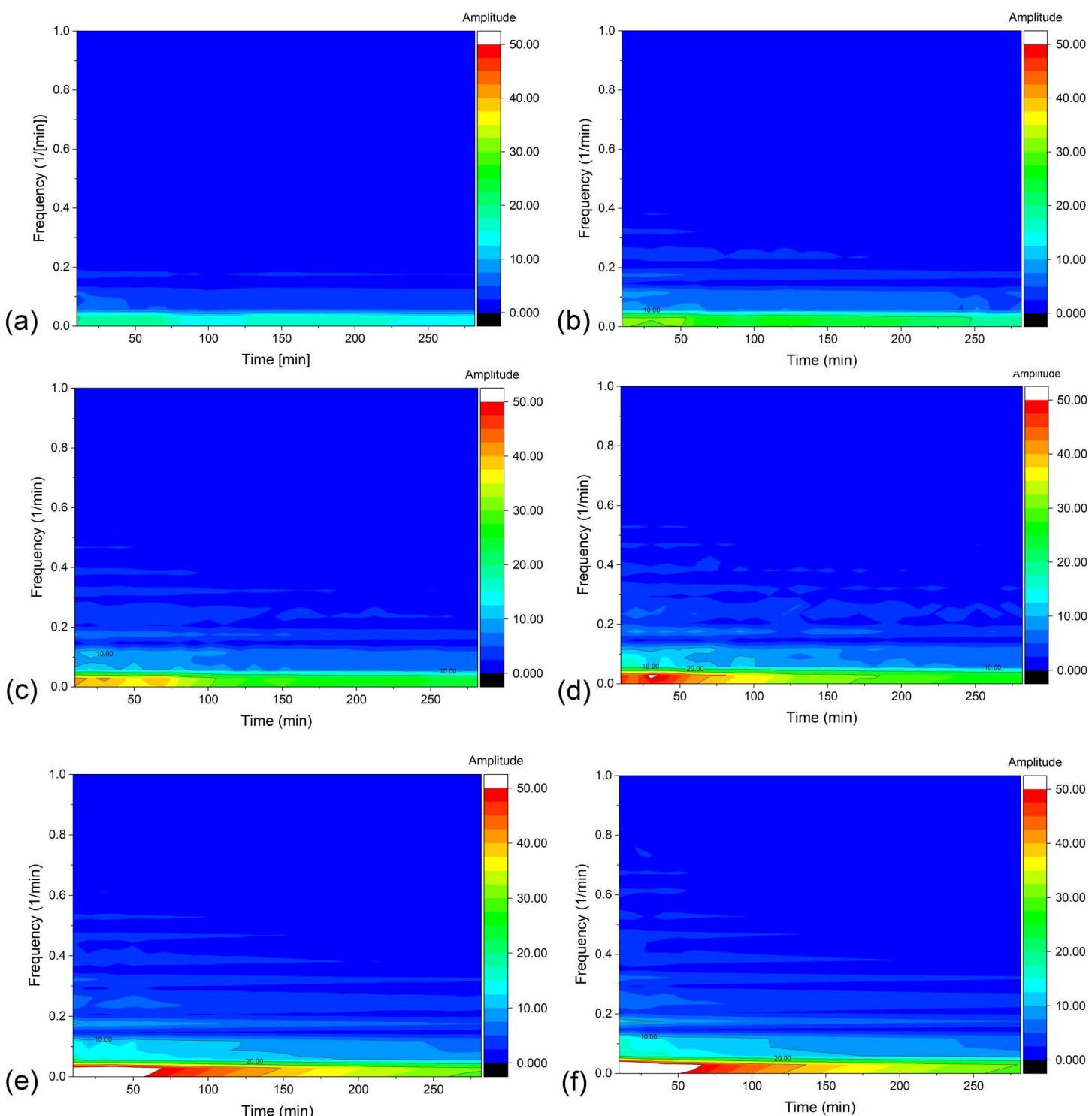

**Fig 5. The distribution of STFT amplitude in time-frequency plane for 20 min. rectangular windows, obtained as a result of STFT for membrane currents, for configuration B of the membrane system and for $(C_h/C_l)_o$ equal to: 100 (a), 500 (b), 1000 (c), 2500 (d), 5000 (e) and 7500 (f).**

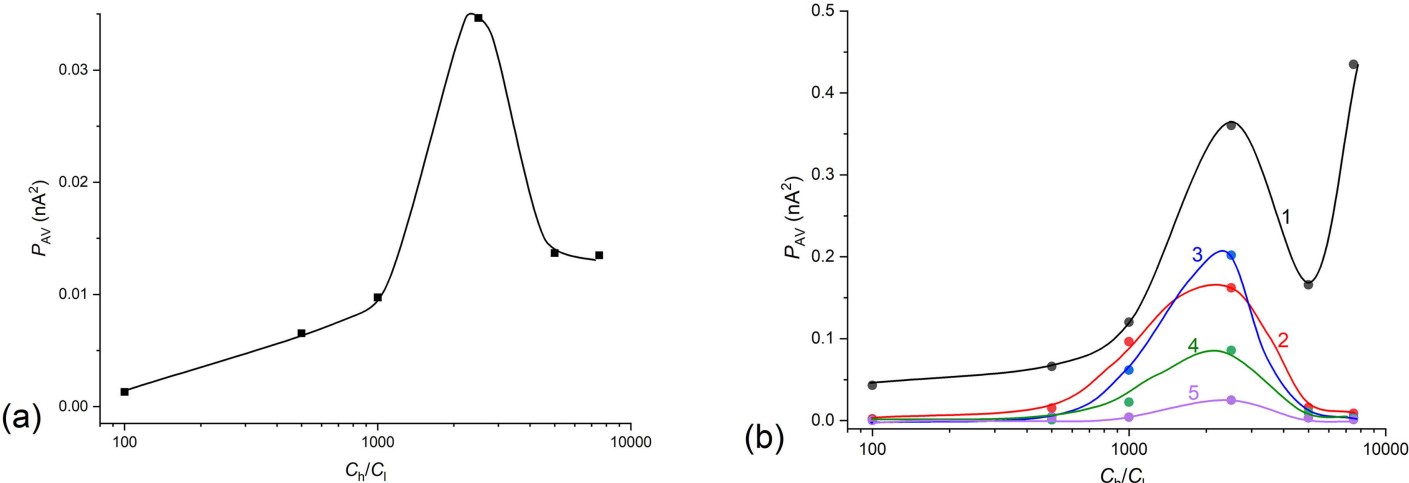

**Fig 6. Average power in the frequency range from 0.05 to 1 min⁻¹ as a function of $(C_h/C_l)_o$ obtained for FFT and for configuration B of the membrane system with bacterial cellulose membrane and NaCl aqueous solutions.** Time intervals of the signal $I(t)$: from 50 to 250 min (a) and 20-minute intervals (b) with the time interval centered at: 20 min (1), 100 min (2), 150 min (3), 200 min (4) and 290 min (5).

50 to 250 minutes) and 9 (STFT analysis of differences of currents in 20-minute rectangular windows with parameters of transformation: Sampling interval = 4s; FFT Length = 512; Window Length = 301 (20 minute window of analysis); Overlap = 150; Window type: Rectangle; Window correction: Amplitude; Option: Amplitude result) present graphs of the Fourier analysis of the time characteristics of the series of current differences for the $(C_h/C_l)_o$ equal to 100 (a), 500 (b), 1000 (c), 2500 (d), 5000 (e) and 7500 (f) respectively.

The graphs in Fig 8 show that there are frequency bands with higher signal power values than the powers in the lower and higher frequency ranges. The increase of $(C_h/C_l)_o$ causes an increase in power amplitudes and for $(C_h/C_l)_o$ greater than 2500 a decrease in power in whole range of frequencies is observed. Besides, a shift of band of maximal powers towards higher frequencies can also be observed with increase of $(C_h/C_l)_o$.

The graphs in Fig 8 do not clearly indicate the dominant frequencies in the presented frequency ranges. Many adjacent peaks with different amplitudes may indicate the random nature of membrane current pulsations, which can be related to the random nature of hydrodynamic instabilities in the chambers of the membrane system. In turn, the graphs in Fig 9 indicate that after removing the decreasing trend line from the membrane current time series, the STFT amplitudes decreased significantly and additional details related to the membrane current pulsations became visible in the graphs for different $(C_h/C_l)_o$. They show that increase of $(C_h/C_l)_o$ causes the increase of STFT amplitudes towards longer times and, at the same time, higher frequencies. However, for $(C_h/C_l)_o$ greater than 2500, areas with higher amplitudes dominate in the first 50 minutes of the experiment, but they disappear with time faster for larger $(C_h/C_l)_o$. In the time-dependent graphs of the membrane currents (Fig 2), this is visible as a rapid vanishing of the current pulsations with time. This may probably be due to relatively smaller changes in concentration differences (concentration gradients) over time compared to smaller $(C_h/C_l)_o$.

For the frequency characteristics shown in Fig 8, the average powers were calculated according to the equation (9) in the frequency range from 0.05 to 1 min⁻¹. The dependencies of calculated averaged powers on $(C_h/C_l)_o$ are presented in Figs 10a (for time range from 50 to 250 min.) and 10b (for 20-minutes intervals), respectively.

As results from Figs 10a and 10b, the maximum average signal power occurs for $(C_h/C_l)_o = 2500$ for all analyzed by FFT time intervals. Shifting of the center of 20-min. time window to later times causes that average power of signals in

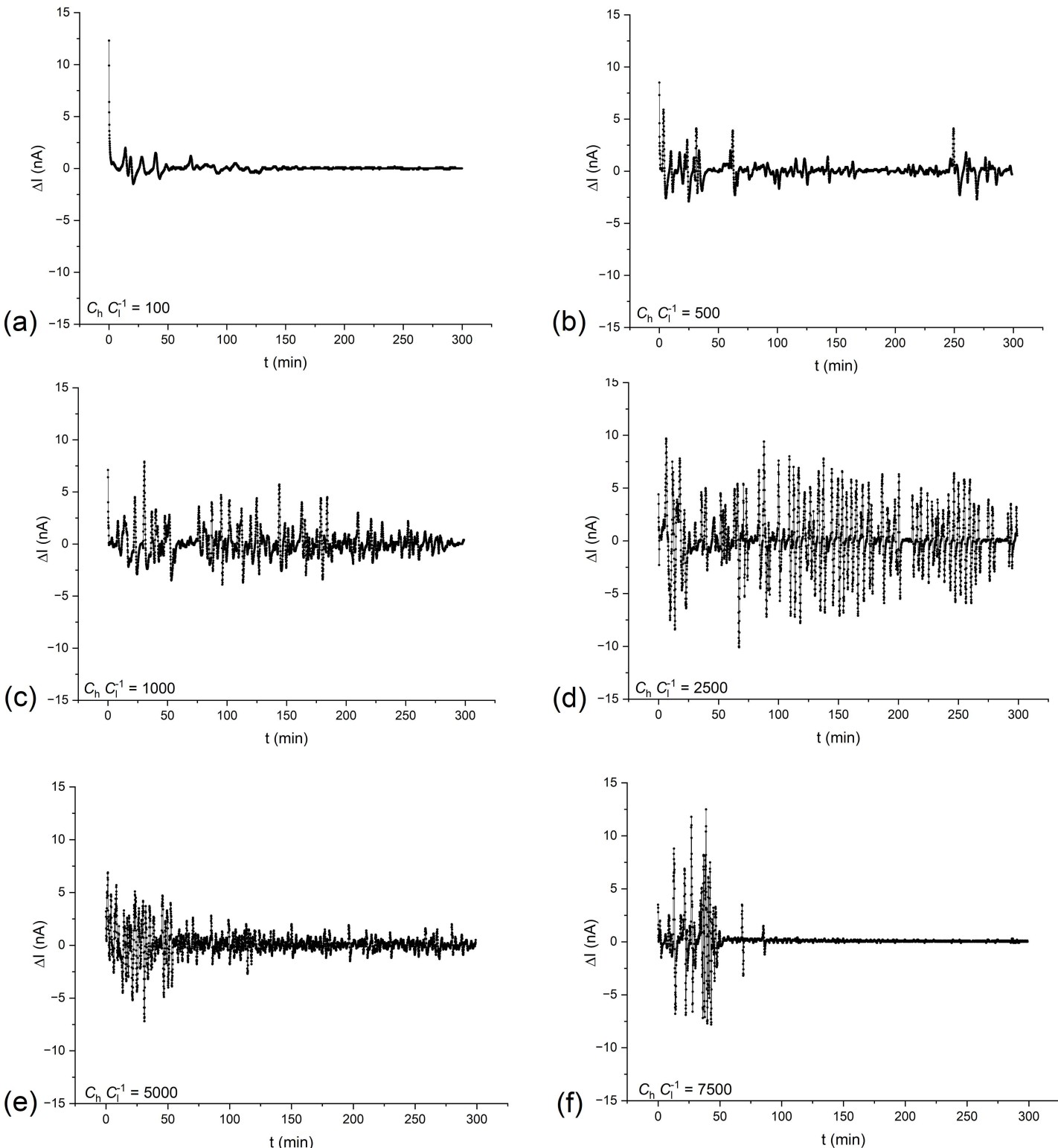

**Fig 7. Time characteristics of differences between elements of the series of current intensities (Δ*I*) for configuration B of the membrane system with bacterial cellulose membrane and for ($C_h/C_l$)$_o$ equal to: 100 (a), 500 (b), 1000 (c), 2500 (d), 5000 (e) and 7500 (f).**

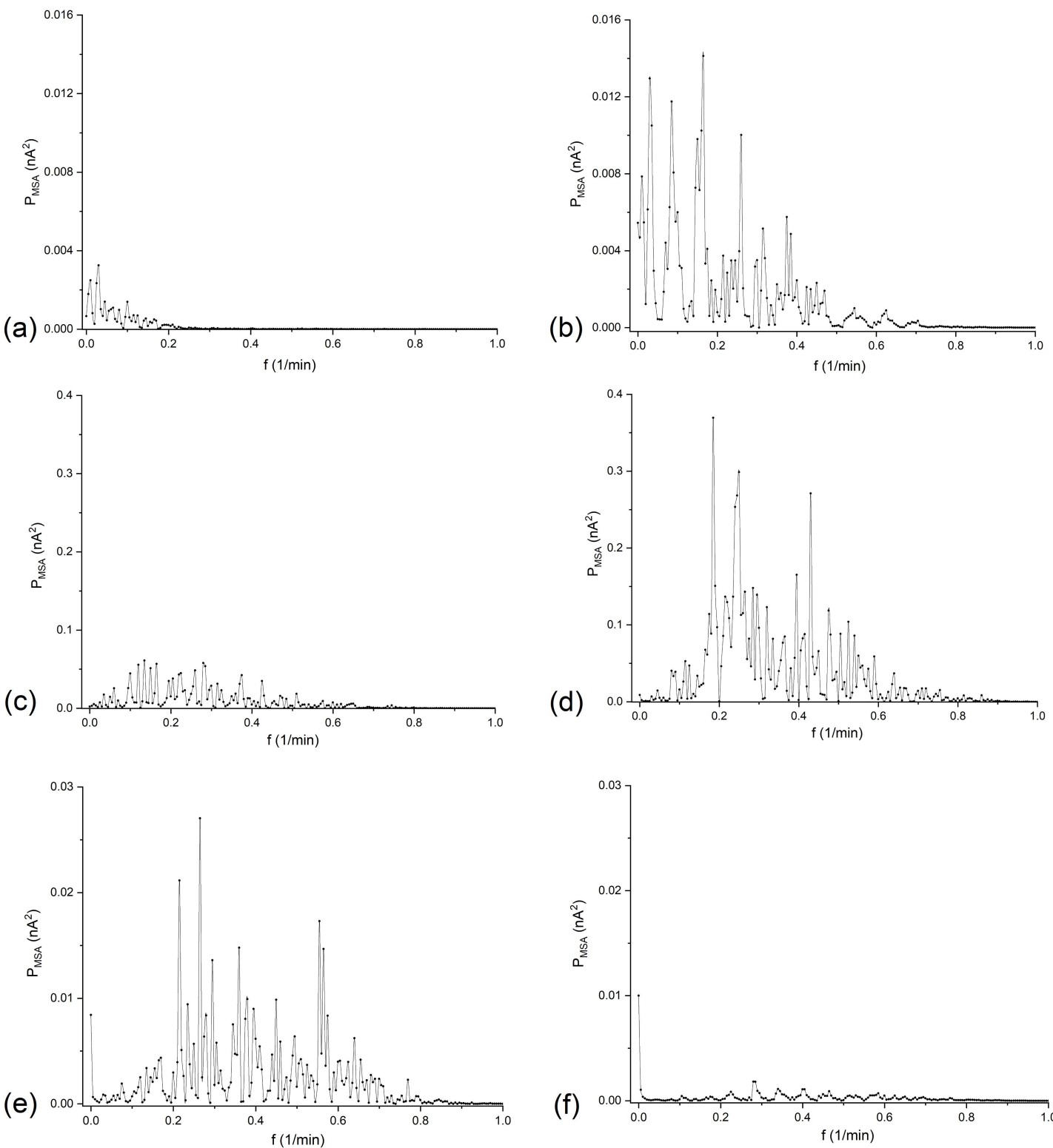

**Fig 8. The dependence of the signal power ($P_{MSA}$) on the frequency, obtained as a result of FFT for differences of currents in configuration B of the membrane system with bacterial cellulose membrane and for time interval 50-250 min., for $(C_h/C_l)_o$ equal to: 100 (a), 500 (b), 1000 (c), 2500 (d), 5000 (e) and 7500 (f), respectively.**

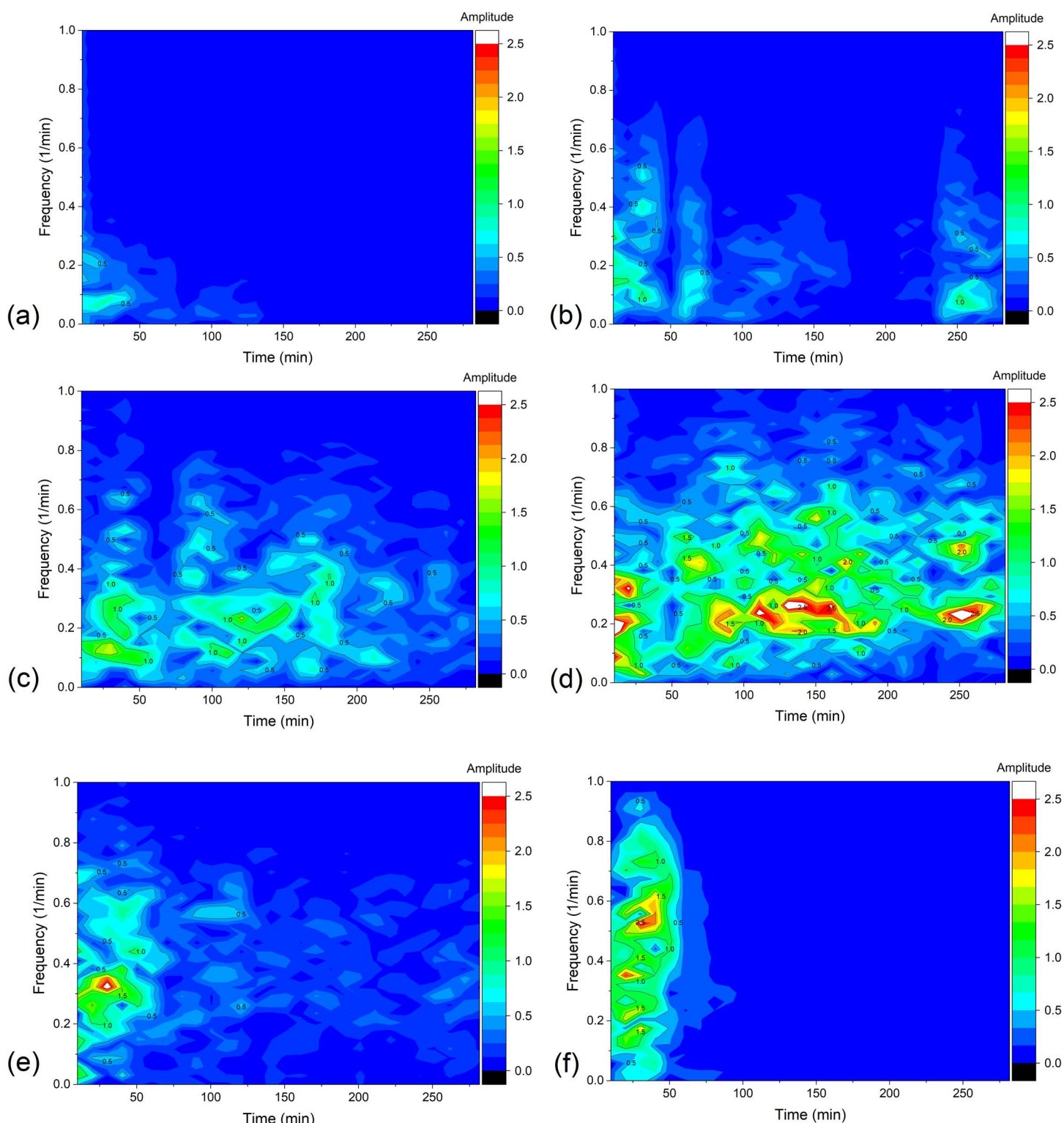

**Fig 9. The distribution of STFT amplitude in time-frequency plane for 20 min. rectangular windows, obtained as a result of STFT for differences of currents, for configuration B of the membrane system with bacterial cellulose membrane and 20-minute rectangular windows, for $(C_h/C_l)_o$ equal to: 100 (a), 500 (b), 1000 (c), 2500 (d), 5000 (e) and 7500 (f).**

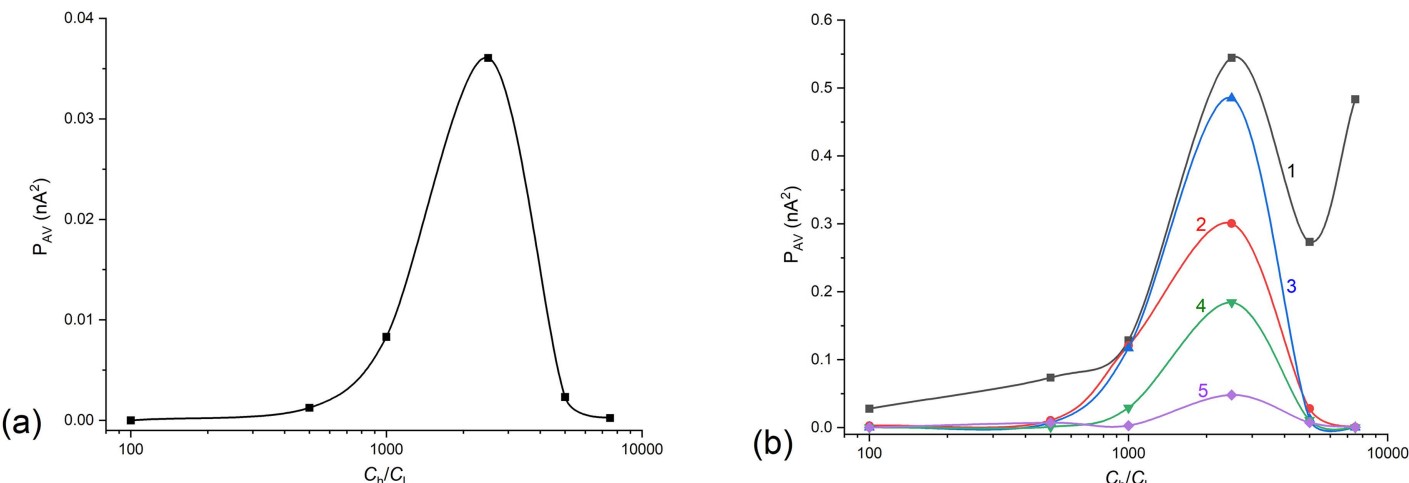

**Fig 10. Average power in the frequency range from 0.05 to 1 min$^{-1}$ as a function of $(C_h/C_l)_o$ for configuration B of the membrane system with bacterial cellulose membrane and NaCl aqueous solutions, obtained as a result of FFT in time intervals of the signal $\Delta I(t)$: from 50 to 250 min (a) and in 20-minute intervals (b) with the center of the time intervals at: 20 min (1), 100 min (2), 150 min (3), 200 min (4) and 290 min (5).**

20-ty minutes intervals (10b) changes over the entire range of $(C_h/C_l)_o$, with a tendency for the average signal power to decrease.

## Conclusions

- The time characteristics of membrane currents in configuration A of the membrane system indicate stable, diffusive reconstruction of CBLs in whole range of $(C_h/C_l)_o$, while for configuration B and $(C_h/C_l)_o \geq 100$, pulsations of membrane currents connected with hydrodynamic instabilities in membrane chambers were observed.

- Membrane currents in steady states, depend on configuration of the membrane system and on $(C_h/C_l)_o$. In steady states and $(C_h/C_l)_o$ greater than 500 higher currents were measured in configuration B than in configuration A.

- FFT analysis of membrane currents shows that the average signal power in the frequency range increases with increasing $(C_h/C_l)_o$, reaching maximum for $(C_h/C_l)_o = 2500$ regardless of analyzed time intervals.

- A procedure that replaces membrane currents in configuration B of the membrane system with the differences between membrane currents separated in time by 1 minute allows for the removal of the decreasing trend of membrane current in time associated with the diffusional reconstruction of CBLs. STFT of transformed current signal better characterizes the observed current pulsations and related to them hydrodynamic instabilities, than the STFT of original current signal.

- The distribution of peaks in the FFT spectrum of the current signal after transformation indicates the random nature of the analyzed hydrodynamic instabilities in the membrane system.

## Supporting information

**S1 Data. Raw data – from measurements and calculations.**
(XLSX)

## Author contributions

**Conceptualization:** Sławomir Grzegorczyn, Andrzej Ślęzak.

**Data curation:** Sławomir Grzegorczyn, Paweł Dolibog, Iwona Dylong.

**Formal analysis:** Sławomir Grzegorczyn, Andrzej Ślęzak.

**Funding acquisition:** Sławomir Grzegorczyn, Iwona Dylong.

**Investigation:** Sławomir Grzegorczyn, Paweł Dolibog.

**Methodology:** Sławomir Grzegorczyn, Andrzej Ślęzak.

**Project administration:** Sławomir Grzegorczyn.

**Resources:** Sławomir Grzegorczyn.

**Software:** Sławomir Grzegorczyn.

**Supervision:** Sławomir Grzegorczyn, Andrzej Ślęzak.

**Validation:** Sławomir Grzegorczyn, Andrzej Ślęzak.

**Visualization:** Sławomir Grzegorczyn, Paweł Dolibog.

**Writing – original draft:** Sławomir Grzegorczyn, Iwona Dylong.

**Writing – review & editing:** Sławomir Grzegorczyn, Paweł Dolibog, Andrzej Ślęzak.

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
