## [Decision Letter · Decision Letter 0]

25 Feb 2026

PONE-D-25-53895Hydrodynamic instabilities in membrane systems with current loading, Fourier analysisPLOS One

Dear Dr. Grzegorczyn,

Thank you for submitting your manuscript to PLOS ONE. After careful consideration, we feel that it has merit but does not fully meet PLOS ONE’s publication criteria as it currently stands. Therefore, we invite you to submit a revised version of the manuscript that addresses the points raised during the review process.

If applicable, we recommend that you deposit your laboratory protocols in protocols.io to enhance the reproducibility of your results. Protocols.io assigns your protocol its own identifier (DOI) so that it can be cited independently in the future. For instructions see: https://journals.plos.org/plosone/s/submission-guidelines#loc-laboratory-protocols. Additionally, PLOS ONE offers an option for publishing peer-reviewed Lab Protocol articles, which describe protocols hosted on protocols.io. Read more information on sharing protocols at . Additionally, PLOS ONE offers an option for publishing peer-reviewed Lab Protocol articles, which describe protocols hosted on protocols.io. Read more information on sharing protocols at https://plos.org/protocols?utm_medium=editorial-email&utm_source=authorletters&utm_campaign=protocols..

We look forward to receiving your revised manuscript.

Kind regards,

Mallikarjuna Reddy Kesama, Ph.D.

Academic Editor

PLOS One

Journal Requirements:

“Iwona Dylong and Sławomir Grzegorczyn

BNW-1-048/K/3/I

Medical University of Silesia

https://sum.edu.pl

NO”

3. We notice that your supplementary figures are uploaded with the file type 'Figure'. Please amend the file type to 'Supporting Information'. Please ensure that each Supporting Information file has a legend listed in the manuscript after the references list.

Reviewers' comments:

Reviewer's Responses to Questions

**Comments to the Author**

1. Is the manuscript technically sound, and do the data support the conclusions?

Reviewer #1: Partly

Reviewer #2: Partly

2. Has the statistical analysis been performed appropriately and rigorously? 

Reviewer #1: No

Reviewer #2: No

3. Have the authors made all data underlying the findings in their manuscript fully available?

Reviewer #1: Yes

Reviewer #2: Yes

4. Is the manuscript presented in an intelligible fashion and written in standard English?

Reviewer #1: Yes

Reviewer #2: No

5. Review Comments to the Author

Reviewer #1: I recommend that the authors provide major clarifications regarding the signal processing methods and the physical interpretation of the non-monotonic dependence of signal power on concentration gradients before this work can be accepted.

Reviewer #2: Comments for authors

The manuscript investigates hydrodynamic instabilities in membrane systems using current measurements and Fourier analysis. The topic fits the technical scope of PLOS ONE and the experiments appear carefully conducted. However, several technical and reporting weaknesses reduce the clarity and reproducibility of the study and should be addressed before the manuscript can be considered technically sound.

Abstract

The abstract is overly descriptive and lacks quantitative clarity. Trends are reported but key numerical outcomes or uncertainty estimates are missing, making it difficult to assess the strength of the findings.

Terminology such as “initial concentration quotient on the membrane” is used repeatedly without clearly defining whether this refers to bulk concentrations or boundary-layer concentrations.

The abstract does not mention experimental replicates or variability, which is important for technical assessment.

Introduction

The introduction provides extensive theoretical background but does not clearly define the specific unresolved technical question addressed by this study.

Literature positioning is limited, with strong reliance on prior work from the same research group and limited comparison with broader recent literature.

The aim statement is descriptive rather than hypothesis-driven, which reduces clarity regarding the study’s objectives.

Materials and Methods

The manuscript does not specify the number of independent experiments, whether figures represent single runs or averages, or the variability between experiments. This limits reproducibility.

No statistical analysis or uncertainty propagation is described for FFT-derived parameters or current measurements.

The FFT methodology is not fully defined. Important details such as windowing, normalization, detrending strategy, and handling of spectral leakage are missing.

The difference method (ΔI) is introduced later in the Results section rather than being fully described in Methods, which affects reproducibility.

The potential influence of instrument loading on the measurements is acknowledged but not quantitatively evaluated.

Results and Discussion

Results and interpretation are strongly mixed, making it difficult to distinguish direct observations from mechanistic interpretation.

No uncertainty estimates or error bars are presented in figures supporting key conclusions, including the reported maximum signal power at Ch/Cl = 2500.

Mechanistic explanations of hydrodynamic instability are inferred from signal behavior without direct validation through flow visualization or complementary measurements.

FFT interpretation appears overextended, as frequency shifts are discussed mechanistically without a quantitative supporting model.

There are figure referencing inconsistencies in the text, suggesting editing issues.

Figures and Figure Legends

Figure legends are not fully self-contained; several lack information about sample size, preprocessing steps, or units for FFT power.

The large number of multi-panel FFT figures reduces clarity; summary metrics could improve readability.

Variability or reproducibility across experiments is not shown in any figure.

Conclusions

The conclusions mainly restate observations rather than demonstrating validated mechanisms.

Some conclusions appear stronger than what is directly supported by the presented data.

Limitations of the study are not discussed, particularly the absence of flow visualization, lack of statistical reporting, and potential measurement-system effects.

6. PLOS authors have the option to publish the peer review history of their article (what does this mean?). If published, this will include your full peer review and any attached files.). If published, this will include your full peer review and any attached files.

.

Reviewer #1: No

Reviewer #2: No

---

## [Author Response · Author response to Decision Letter 1]

8 Apr 2026

Responses to reviewers' comments

Manuscript Title: Hydrodynamic instabilities in membrane systems with current loading, Fourier analysis

Manuscript ID: PONE-D-25-53895

Review 1:

1. Differentiation between Convection Types: The authors suggest that ”current loading” introduces additional concentration disturbances compared to voltage measurements. However, the manuscript does not clearly distinguish between hydrodynamic instabilities caused purely by gravitational density gradients (Rayleigh-Benard type) and those potentially induced or exacerbated by the current flow itself (electro-convection).

Question: Did the authors perform control experiments with varying external resistances or zero-current conditions to isolate the ”current load” effect from the density effect?

AD. 1.

In the case of high externally forced currents (by additional EMF - active current forcing), hydrodynamic instabilities in solutions may occur. However, the case under consideration concerns the passive generation of currents in membrane systems. This means that the currents are generated by differences in electrolyte concentrations at the surfaces of electrodes connected in the membrane system to a nanoammeter (low external resistance) used to measure these currents. Therefore, the values of these currents are not large enough to cause hydrodynamic instabilities on their own. However, they can be used by measuring their changes over time to monitor processes occurring in the membrane system (changes in electrolyte concentrations, formation of CBLs near the membranes, or the appearance and evolution in time of hydrodynamic instabilities related to natural convection in the gravitational field and caused by the density gradients appearing in CBLs opposite to the gravitational field vector). For this reason, the formulations regarding the influence of these currents on hydrodynamic instabilities are too strong, therefore the text of the article was re-examined and such suggestions were removed from the text of the article. Our experiments did not concern determining the influence of the current flowing in the membrane system on hydrodynamic instabilities, but the use of measured current to monitor processes in the membrane system. In the case under consideration, the observed currents are a consequence of changes in concentrations in the membrane system and not the cause of their disturbance. Referring to the reviewer's question, the carried out experiment cannot be considered as allowing the separation of the current effect from the density effect of hydrodynamic instabilities. Additionally, no tests were performed with variable resistances connected in series to the nanoammeter. The idea is interesting, however, due to the passive nature of current generation in the membrane system, it would probably lead to a reduction in current intensities under analogous conditions of the membrane system, without a significant impact on instabilities related to density effects.

2. Detrending Algorithm and Frequency Response: The data preparation for FFT involves a differential algorithm defined as ΔI(ti) = I(ti) − I(ti + 1 min). This operation effectively acts as a high-pass filter.

Question: How does the specific choice of a 1-minute time lag impact the detection of lower-frequency pulsation modes? Was a sensitivity analysis performed with different time lags (e.g., 2 or 5 minutes) to ensure that the observed peak power at Ch/Cl = 2500 is not an artifact of the filter width matching a specific process time-scale?

AD. 2.

The selection of the time difference (�t=1min) was preceded by an analysis of the behavior of the differential signal for various time differences. The new differential signals allowed the decreasing trend to be eliminated, but after transformations remained pulsations of various amplitudes and this same frequencies. Due to the analysis of current signals (their changes over time) in the tested range of initial conditions of the membrane system (pulsation periods were greater than 1 minute), the signal shift time in the differential algorithm was chosen to be 1 minute. The analysis of selected signals with different shift times (up to 5 minutes) shows that this does not disturb the signal pulsations (i.e. their frequency character) but only affects their amplitudes. Therefore, the main criterion for selecting the shift time were the observed pulsations (their frequencies) of the recorded membrane currents. As can be seen from the Fourier analysis, the main frequencies of the membrane currents detected in the spectrum are in the range of values below 1 min-1 (main band between 0.05 and 0.06 min-1). The example graphs after applying a 2-minute and 5-minute shift (compared to 1 minute), (for configuration B and (Ch/Cl)o =2500) are shown in the figure on the next page. (Perhaps using greater signal shift times could result in "clipping" of the parts of signal at higher frequencies – around 0.5 min-1, occurring in signals with higher (Ch/Cl)o values).

3. Statistical Robustness: The figures currently present single curves for specific concentration quotients. Question: How many experimental repetitions were conducted for each Ch/Cl point in figure? The manuscript should include error bars or standard deviations for the ”Average Power” presented in Figure 6 and Figure 10 to demonstrate that the peak

1 at 2500 is statistically significant. All rest figures from figure 2 to figure 10 also need error bars.

AD. 3.

The experiment was aimed at measuring a series of measurements for different initial conditions (different (Ch/Cl)o). Measurements performed for the same conditions (both for voltages between electrodes and currents flowing in the membrane system) show that the standard error of measurements of the initial current do not exceeded 5%, while the standard error of measurements of currents in steady states do not exceeded 12%. The estimations of the measurement error of the current intensities in the initial and in the steady states were performed five times for the selected initial quotients of NaCl concentrations in the chambers (Ch/Cl)o. An error of this order can be plotted on the graphs of initial and steady-state states in configurations A and B as a function of (Ch/Cl)o (Fig. 3). As for the curves (Power as a function of (Ch/Cl)o- Figs. 6 and 10), the calculations were performed for sample single I(t) runs and therefore the measurement accuracy cannot be taken into account in these graphs.

Fig. R1 Time characteristics of differences between elements of the series of current intensities (�I) for configuration B of the membrane system and for the initial NaCl concentrations quotients Ch/Cl = 2500 on the membrane, and for �t equal to: 1min (a), 2min (b) and 5min (c).

(in file: Response to Reviewers.docx)

4. Physical Mechanism of the Power Decrease: The results indicate a maximum signal power at Ch/Cl = 2500, followed by a decrease at higher quotients (5000, 7500). Classical theory suggests that higher density gradients usually lead to more intense turbulence (higher Rayleigh numbers). Question: What is the physical mechanism causing the suppression of instability intensity at the highest concentrations? Is this related to a saturation of the membrane transport capacity, a change in solution viscosity, or another hydrodynamic factor?

AD. 4.

In the observed hydrodynamic instabilities, increasing the initial concentration gradients through the membrane causes larger density gradients to appear in the CBLs over time, resulting in more intense hydrodynamic instabilities in configuration B of the membrane system. However, when measuring the temporal changes in membrane voltages for various membranes, we observed large voltage pulsations for (Ch/Cl)o greater than or equal to 5000, which were strongly damped after some time of appearance. As a result of hydrodynamic instabilities, intense stirring of solutions in the chambers occurs in an area larger than the CBLs. Increase of the initial concentration gradient across the membrane (increase of (Ch/Cl)o) causes greater electrolyte transport through the membrane, resulting in a faster increase of concentration gradients (and solution density) in CBLs and higher intensities of hydrodynamic instabilities (pulsations of both observed voltages and currents with higher amplitude and frequency). However, at high (Ch/Cl)o values (as was noted by the reviewer), there may be a saturation effect of electrolyte transport through the membrane. Besides, a faster equalization of concentrations in the chambers (at the level of higher concentrations) may occur. These effects may finally lead to a faster decrease of concentration gradients (as well as density gradients) in the chambers which may lead to faster damping of instability (and thus pulsation of observed electrical parameters). Previous analyses of concentration changes at membrane surfaces in time for the stable configuration after turning off mechanical stirring of solutions indicate that the largest changes in these concentrations occur within a few minutes after turning off mechanical stirring. After this time, membrane transport is significantly limited by the formation of CBLs. The emerging hydrodynamic instabilities in CBLs caused by gravity also disturb the concentrations at the membrane surfaces, leading to temporary increase in membrane transport, or rather, pulsatile changes in these concentrations. This is due to the opposing effects of diffusion through the membrane and hydrodynamic instabilities on changes in concentration at the membrane surfaces. Perhaps, for large (Ch/Cl)o, these relative concentration disturbances at the membrane are smaller compared to smaller (Ch/Cl)o, which may result in a lower possibilities of reproducing density gradients leveled by hydrodynamic instabilities. As a result, a damping effect on hydrodynamic instabilities sustained by density gradients in the chambers solutions can be observed. Due to the fact that the signal power in the Fourier transform is connected with the pulsation intensity, it can be stated that a greater damping of current pulsations over time causes a decrease in the average power in the considered frequency range. In the case of the twenty-minute interval with the interval centered at the 20th minute, due to the high intensity of pulsation in the initial phase of instability, high average powers in this time interval are observed also for (Ch/Cl)o greater than (or equal to) 5000. These processes and their potential influence on the nature of instability (current pulsations) were taken into account in explaining the dependence of the average power as a function of (Ch/Cl)o.

5. Steady-State vs. Dynamic Analysis: The authors note that for Ch/Cl > 25, steady state currents in Configuration B remain higher than in A due to CBL ”blurring.” However, for very high concentrations, the difference between steady states appears to decrease. Question: Does this convergence of steady-state values at high concentrations correlate with the decrease in FFT signal power observed above Ch/Cl = 2500? A unified explanation linking the steady-state current data and the FFT dynamic analysis is currently missing.

AD. 5.

Indeed, as was noted by the reviewer, the difference between the steady states for configurations A and B of the membrane system – shown in Fig. 3 for large (Ch/Cl)o (above 750) decreases compared to smaller (Ch/Cl)o values (between 100 and 750). The likely cause is (similarly to the observed pulsation intensities and the associated average signal powers obtained in Fourier analysis) the effect of damping the instability over time. It can be suspected that the time after which the instabilities are suppressed due to the effective disappearance of density (concentration) gradients is shorter for higher (Ch/Cl)o, and during measurements up to 300 minutes it is visible for (Ch/Cl)o greater than (or equal to) 5000. It is true that for smaller (Ch/Cl)o smaller amplitudes of hydrodynamic instabilities are observed (smaller amplitudes and frequencies of current or voltage pulsations), but due to their longer duration they may result in larger differences between the steady states for configurations A and B compared to larger (Ch/Cl)o.

Review 2:

Abstract

The abstract is overly descriptive and lacks quantitative clarity. Trends are reported but key numerical outcomes or uncertainty estimates are missing, making it difficult to assess the strength of the findings. Terminology such as “initial concentration quotient on the membrane” is used repeatedly without clearly defining whether this refers to bulk concentrations or boundary-layer concentrations.

The abstract does not mention experimental replicates or variability, which is important for technical assessment.

Ad. (Abstract):

Analyzing the hydrodynamic instabilities in configuration B of the membrane system, by means of the Fast Fourier Transform in the range of time from 50 to 250 min. we can state that the average power of the current signal nonlinearly depends on the (Ch/Cl)o (initial bulk solutions in chambers, homogeneous during mechanical stirring of solutions) with maximum at (Ch/Cl)o = 2500. The transformation of current signal (difference of currents with shift time equal 1 minute) allows to eliminate the decreasing trend line connected with rebuilding of CBLs near membrane. The Short Time Fourier Transform (STFT) of transformed current signal with 20 minute rectangular windows shows increasing amplitudes with increasing (Ch/Cl)o up to 2500 (in later times and higher frequencies). For (Ch/Cl)o greater than 2500 lower amplitudes of transformed signals for higher frequency and longer times were observed, what can be associated with higher dumping of hydrodynamic instabilities in the membrane system. Calculation of average power in 20min intervals showed similar dependencies of average power as a function of (Ch/Cl)o in all intervals, as was the case for the signal in the range of time from 50 to 250 min. The abstract and methodology have been revised.

Introduction

The introduction provides extensive theoretical background but does not clearly define the specific unresolved technical question addressed by this study. Literature positioning is limited, with strong reliance on prior work from the same research group and limited comparison with broader recent literature. The aim statement is descriptive rather than hypothesis-driven, which reduces clarity regarding the study’s objectives.

Ad. (Introduction):

In the introduction, we have improved the description of the article's objectives and extended the signal analysis based on the Short Time Fourier Transform (STFT) method. The Fast Fourier Transform (FFT) and Short Time Fourier Transform (STFT) of the current signal and its "differential" transformation were used to demonstrate the complex nature of the distribution of signal power maxima and amplitude distribution. The Fourier transform of membrane currents could indicate the random (stochastic) nature of hydrodynamic instabilities related to natural convection, induced by sufficiently large solution density gradients near the membrane surface.

Materials and Methods

The manuscript does not specify the number of independent experiments, whether figures represent single runs or averages, or the variability between experiments. This limits reproducibility. No statistical analysis or uncertainty propagation is described for FFT-derived parameters or current measurements. The FFT methodology is not fully defined. Important details such as windowing, normalization, detrending strategy, and handling of spectral leakage are missing. The difference method (ΔI) is introduced later in the Results section rather than being fully described in Methods, which affects reproducibility. The potential influence of instrument loading on the measurements is acknowledged but not quantitatively evaluated.

Ad. (Materials and Methods):

The measurements were performed as a series for various (Ch/Cl)o. Independent measurements showed that the standard error of measurements of the initial current do not exceeded 5%, while the standard error of measurements of currents in steady states do no

---

## [Decision Letter · Decision Letter 1]

19 Apr 2026

Hydrodynamic instabilities in membrane systems with current loading, Fourier analysis

PONE-D-25-53895R1

Dear Dr. Slawomir Grzegorczyn,

We’re pleased to inform you that your manuscript has been judged scientifically suitable for publication and will be formally accepted for publication once it meets all outstanding technical requirements.

An invoice will be generated when your article is formally accepted. Please note, if your institution has a publishing partnership with PLOS and your article meets the relevant criteria, all or part of your publication costs will be covered. Please make sure your user information is up-to-date by logging into Editorial Manager at Editorial Manager® and clicking the ‘Update My Information' link at the top of the page. For questions related to billing, please contact  and clicking the ‘Update My Information' link at the top of the page. For questions related to billing, please contact billing support..

Kind regards,

Mallikarjuna Reddy Kesama, Ph.D.

Academic Editor

PLOS One

Additional Editor Comments (optional):

Accept

Reviewers' comments:

Reviewer's Responses to Questions

**Comments to the Author**

1. If the authors have adequately addressed your comments raised in a previous round of review and you feel that this manuscript is now acceptable for publication, you may indicate that here to bypass the “Comments to the Author” section, enter your conflict of interest statement in the “Confidential to Editor” section, and submit your "Accept" recommendation.

Reviewer #1: All comments have been addressed

Reviewer #2: All comments have been addressed

2. Is the manuscript technically sound, and do the data support the conclusions?

Reviewer #1: Yes

Reviewer #2: Yes

3. Has the statistical analysis been performed appropriately and rigorously? 

Reviewer #1: No

Reviewer #2: Yes

4. Have the authors made all data underlying the findings in their manuscript fully available?

Reviewer #1: Yes

Reviewer #2: Yes

5. Is the manuscript presented in an intelligible fashion and written in standard English?

Reviewer #1: Yes

Reviewer #2: Yes

6. Review Comments to the Author

Reviewer #1: (No Response)

Reviewer #2: All modifications and revisions were made. The authors replied satisfactorily to my comments. Thank you

7. PLOS authors have the option to publish the peer review history of their article (what does this mean?). If published, this will include your full peer review and any attached files.). If published, this will include your full peer review and any attached files.

.

Reviewer #1: **Yes:** Wenzheng ShiWenzheng Shi

Reviewer #2: No

---

## [Editor Report · Acceptance letter]

PONE-D-25-53895R1

PLOS One

Dear Dr. Grzegorczyn,

I'm pleased to inform you that your manuscript has been deemed suitable for publication in PLOS One. Congratulations! Your manuscript is now being handed over to our production team.

Kind regards,

on behalf of

Dr. Mallikarjuna Reddy Kesama

Academic Editor

PLOS One